

**Representation of the Autoconversion from Cloud to Rain Using a**
**Weighted Ensemble Approach**
**Jinfang Yin[1]\*, Xudong Liang[1], Hong Wang[2], Haile Xue[1]**
1 State Key Laboratory of Severe Weather (LaSW), Chinese Academy of
Meteorological Sciences (CAMS), Beijing 100081, China
2 Guangzhou Institute of Tropical and Marine Meteorology, China Meteorological
Administration (CMA), Guangzhou 510080, China
*Corresponding to*: Jinfang Yin (yinjf@cma.gov.cn)





**Abstract.** Cloud and precipitation processes remain among the largest sources of
uncertainties in weather and climate modeling, and considerable attention has been
paid to improve the representation of the cloud and precipitation processes in
numerical models in the last several decades. In this study, we develop a weighted
ensemble (named as EN) scheme by employing several widely used autoconversion
(ATC) schemes to represent the ATC from cloud water to rainwater. One unique
feature of the EN approach is that ATC rate is a weighted mean value based on the
calculations from several ATC schemes within a microphysics scheme with a
negligible increase of computation cost. The EN scheme is compared with the several
commonly used ATC schemes by performing a real case simulations. In terms of
accumulated rainfall and extreme hourly rainfall rate, the EN scheme provides better
simulations than that by using the single Berry-Reinhardt scheme which was
originally used in the Thompson scheme. It is worth emphasizing, in the present study,
we only pay our attention to the ATC process from cloud water into rainwater with the
purpose to improve the modeling of the extreme rainfall events over southern China.
Actually, any (source/sink) term in a cloud microphysics scheme can be dealt with the
same approach. The ensemble method proposed herein appears to have important
implications for developing cloud microphysics schemes in numerical models,
especially for the models with variable grid resolution, which would be expected to
improve of the representation of cloud microphysical processes in the weather and
climate models.





## 1 Introduction

Cloud and precipitation processes and associated feedbacks have been confirmed to cause the largest uncertainties in weather and climate modeling by the Intergovernmental Panel on Climate Change (IPCC) (Houghton et al., 2001). Owing to the complex microphysical processes in clouds and their interactions with dynamical and thermodynamic processes, considerable attention has been devoted to developing cloud microphysics schemes in the numerical weather and climate models in the last several decades, which is summarized in several review articales (e.g., Grabowski et al., 2019; Khain et al., 2015; Morrison et al., 2020). Because of fundamental gaps in the knowledge of cloud microphysics, however, there are still a large number of empirical values derived and assumptions in microphysics schemes based on limited observations, even from numerical simulations (Tapiador et al., 2019). As a result, simulations are quite sensitive to microphysical parameter settings (Falk et al., 2019; Freeman et al., 2019; Gilmore et al., 2004), and thus obvious differences occur frequently from different simulations due to the poor representation of the empirical values and assumptions (Lei et al., 2020; White et al., 2017).

Collision–coalescence between cloud droplets forming riandrops is named as the autoconversion (ATC) , which is a significant microphysical process in warm clouds. Therefore, representation of the ATC from cloud water to rainwater is a key aspect of cloud microphysical parameterization. Firstly, raindrop is initiated by ATC process in warm clouds, which plays a significant role in the onset of a rainfall event. Besides, ATC process has important influence on cloud microphysical properties by bridging aerosols, cloud droplets, and raindrops (White et al.,



2017). Additionally, local circulation may be modified to a certain extent due to falling down
of the initialized raindrops because of terminal velocity of raindrop (Doswell, 2001).
Moreover, changes in the rate of ACT had some effect on the lower-tropospheric radiative
flux divergence (Grabowski et al., 1999). Consequently, an appropriate representation of the
ATC process is helpful for our understanding of cloud micro- and macro-properties, as well as
precipitation processes.
Over the last several decades, much attention has been devoted to establishing ATC
schemes in atmospheric numerical models, and efforts are under way to create accurate and
computationally efficient ATC schemes. Kessler (1969) pioneered a simple scheme in which
the ATC rate was connected to cloud water content (CWC), and the scheme has been widely
used in bulk microphysics schemes (e.g., Chen and Sun, 2002; Dudhia, 1989; Ghosh and Jonas,
1999; Rutledge and Hobbs, 1984). As an alternate way, Berry (1968) established an more
physical formulation in which not only CWC was considered but also cloud droplet number
concentration ($N_c$) and spectral shape parameter of cloud droplet size distribution. The Berry
scheme was featured by estimating the time $t$ required for the sixth-moment diameter of the
spectral density to reach 80 μm by droplet coalescence, and Simpson and Wiggert (1969)
increased the sixth-moment diameter to 100 μm. Ghosh and Jonas (1999) proposed a scheme
by combining the advantages of the Kessler and Berry schemes, which allow the use of the
simple linear Kessler-type expression and incorporating the effects of different cloud types. On
the other hand, several model-derived empirical schemes was established on the basis of
sophisticated microphysical simulations (Berry and Reinhardt, 1974; Franklin, 2008;
Khairoutdinov and Kogan, 2000; Lee and Baik, 2017). Recently, Some studies (e.g., Franklin,





2008; Li et al., 2019; Onishi et al., 2015; Seifert et al., 2010) the effect of turbulence on ATC
have been taken into account. Naeger et al. (2020) proposed that a neglect of turbulence
influence within a ATC scheme resulted in very weak condensational and collisional growth
processes, and thus underpredicted the contribution of warm rain processes to the surface
precipitation. More recently, multi-moment schemes were explored, which appeared to
improve precipitation simulation in a certain extent (Kogan and Ovchinnikov, 2019).

To date, numerous of ATC schemes have been established (Beheng, 1994; Berry, 1968;

Berry and Reinhardt, 1974; Caro et al., 2004; Franklin, 2008; Kessler, 1969; Kogan and
Ovchinnikov, 2019; Lee and Baik, 2017; Lin et al., 2002; Liu and Daum, 2004; Liu et al., 2006;
Manton and Cotton, 1977a; Seifert and Beheng, 2001; Wood et al., 2002; Yin et al., 2015). As
were noted in previous studies (Gilmore and Straka, 2008; Hsieh et al., 2009; Liu et al., 2006;
Xiao et al., 2020; Yin et al., 2015), ATC rates predicted by different schemes can differ by
several orders of magnitude for a given CWC. Many previous studies have shown that ATC
rates are often overestimated/underestimated by those ATC schemes. For instance, Cotton
(1972) pointed out that the Kessler's formulation produced the largest error at smaller CWCs,
and Berry's formulation consistently resulted in a low rain rate low in the simulated clouds.
Iacobellis and Somerville (2006) proposed that the Manton-Cotton parameterization (Manton
and Cotton, 1977b) produced much larger values of liquid water path (LWP) than
measurements both by satellites and surface-based at the Atmospheric Radiation Measurement
(ARM) Program's  Southern U.S. Great Plains site. Silverman and Glass (1973) addressed
that the Cotton (1972) scheme resulted in a peak cloud water content that was in lowest in value,
occurred earliest in time, and occurred at the lowest height in clouds, compared to those of the



Kessler (1969), Berry (1968), and Simpson-Wiggert (1969) schemes. However, Flatøy (1992)
stated that Sundqvist's (Sundqvist et al., 1989) and Kessler's (Kessler, 1969) schemes gave
comparable results when used a suitable choice of parameters. To the best of our knowledge,
however, there is no one ATC parameterization scheme able to provide good results at all times
so far, and much effort is necessary for further development of the ATC parameterization
(Michibata and Takemura, 2015).
As noted by Morrison et al. (2020), one of the most serious issues of treating
microphysics in weather and climate models is the uncertainties in the microphysical process
rates owing to fundamental gaps in the knowledge of cloud physics. Posselt et al. (2019)
proposed that changes in cloud microphysical parameters produced the same order of
magnitude change in model output as did changes to initial conditions, and thus it was
important to constraint uncertainties in cloud microphysical processes if possible. Wellmann
et al. (2020) also pointed out that model dynamical and microphysical properties were
sensitive to both the environmental and microphysical uncertainties, and the latter resulted in
larger uncertainties in the output of integrated hydrometeor mass contents and precipitation
variables.
There is still poor representation of ATC process in weather and climate models, and the
potential uncertainties are non-negligible in the ATC schemes (Michibata and Takemura,
2015), and continued advancement of parameterizations are require greater knowledge of the
underlying physical processes in order to reduce the uncertainties, including from laboratory
studies, cloud observations, and detailed process modeling (Randall et al., 2019). Most
importantly, representing cloud processes consistently across multi-scales models with an





empirical scheme appears to be one of the major challenges in the cloud parameterizations
(Randall et al., 2019). To fill this gap, the objective of this paper is to address how to reduce the
negative effects of inherent uncertainties in the ATC (from cloud water to rainwater)
parameterization within a cloud microphysics scheme to make the weather and climate models
behave realistically. To achieve this goal, we design a weighted ensemble (herein abbreviated
as EN) scheme to represent the ATC process by employing several widely used ATC schemes
within a cloud microphysics scheme.
This paper is organized as follows. An overview of the selected ATC schemes is presented
in Section 2. Section 3 describes the approach of ensemble scheme. The Weather Research and
Forecasting (WRF) model configuration and experiment settings are given in Section 4.
Simulated results of an extreme rainfall event are presented in Section 5. Finally, conclusion
and discussions are given in Section 6.

## 2 Overview of the selected autoconversion schemes

In the present study, four widely used ATC schemes are selected, including Kessler (1969)
(KE) scheme, Berry and Reinhardt (1974) (BR) scheme, Khairoutdinov and Kogan (2000)
(KK) scheme, and Liu et al. (2006) (LD) scheme. Depending on properties of the "bulk"
microphysics schemes, the KE scheme is a one-moment scheme, and the BR and KK are
double-moment schemes. The LD scheme provides a generalized expression with smooth
transition in the vicinity of the ATC threshold, which is featured by eliminating unnecessary
assumptions inherent in the existing Kessler-type parameterizations. It should be noted it is
still troublesome to justify in recommending one of the ATC schemes over the other, although





those schemes have been extensively tested and widely used in the previous studies (Gilmore
and Straka, 2008; Jing et al., 2019; Michibata and Takemura, 2015; White et al., 2017).
**2.1 Kessler (KE) scheme**

Kessler (1969) pioneered a simple expression in which ATC rate is related to CWC. The

KE scheme has been widely used in cloud-related processes in weather and climate numerical
models due to its simplicity. The ATC rate from cloud water to rainwater is expressed as
$$P_{ATC-KE}[\mathrm{kg\,kg^{-1}\,s^{-1}}] = r_a a(q_c - q_0)H(q_c - q_0) \quad \begin{cases} q_c - q_0 \geq 0, H(q_c - q_0) = 1, \\ q_c - q_0 < 0, H(q_c - q_0) = 0. \end{cases} \tag{1}$$

where $a$ =0.001 s$^{-1}$ is a time constant, $H$ is the Heaviside function, $q_c$ is CWC in unit of kg
m$^{-3}$, and $r_a$ is air density. The threshold $q_0$ is the minimum CWC below which there is no
ATC from cloud water to rainwater (Fig. 1a). Owing to the simple and linear expression, the
KE scheme is computationally straightforward to implement in numerical models. However,
the major limitation of the KE scheme results in its inability to identify different conditions
such as maritime and continental clouds (Ghosh and Jonas, 1999). Besides, it is impossible to
obtain the thresholds directly used in the scheme from observations at present, while cloud
microphysical processes are sensitive to the thresholds (Plsselt et al., 2019). A modified
Kessler scheme was proposed by Yin et al. (2015) in which $q_0$ is diagnosed as a function of
altitude by using a CWC-height relationship which was derived from CloudSat observations.
In fact, different values of $q_0$ were chosen by various studies. For instance, a value of 0.5 g
m$^{-3}$ is given in Kessler's (1969), Reisner (1998), and Schultz (1995). Thompson (2004)
reduced to a small value of 0.35 g m$^{-3}$. Kong and Yau (1997) and Tao and Simpson (1993)
gave a value of 2 g kg$^{-1}$, while a small value of 0.7 g kg$^{-1}$ was assigned in Chen and Sun



(2002). In this work, the same value of 0.5 g m⁻³ as that assigned in Kessler's (1969) is
chosen.

**2.2 Berry-Reinhardt (BR) scheme**

Berry and Reinhardt (1974) proposed an physical formulation to represent ATC process in
clouds, which is given by

$$P_{ATC-BR}[\text{kg kg}^{-1}\,\text{s}^{-1}] = \frac{2.7\times10^{-2}\,r_w q_c\left[\frac{1}{16}\times10^{20}D_{mean}^4(1+m)^{-0.5}-0.4\right]}{\frac{3.7}{r_a q_c}\left[0.5\times10^6 D_{mean}(1+m)^{-1/6}-7.5\right]^{-1}}. \qquad (2)$$

Here, $m$ represents shape parameter of a gamma distribution, $r_w$ is liquid water density. $D_{mean}$
is the mean diameter (unit in m) of the total cloud droplets, which is computed from

$$D_{mean} = \left(\frac{6q_c}{pr_w N_c}\right)^{1/3}. \qquad (3)$$

Here, $p$ is the circumference ratio. Compared to KE, the BR scheme has treated the process
more rigorously (Ghosh and Jonas, 1999). It should be noted that ATC rates given by BR are
quite sensitive to $N_c$ (Fig. 1b).

**2.3 Khairoutdinov-Kogan (KK) scheme**

Khairoutdinov and Kogan (2000) proposed a computationally efficient and relatively
simple scheme, which aims at large-eddy simulation (LES). One of the advantages is that there
is no need to define a threshold, and this scheme has been broadly used in numerical models
(e.g., Morrison et al., 2009). The ATC rate is given by

$$P_{ATC\text{-}KK}[\text{kg kg}^{-1}\,\text{s}^{-1}] = 1350 q_c^{2.47}(N_c\times10^{-6})^{-1.79}. \qquad (4)$$

The KK scheme uses a simple power law expression based on bin microphysical calculations.
The simple expression is a key advantage of the KK scheme, which makes it possible to



analytically integrate the microphysical process rates over a probability density function
(Griffin and Larson, 2013). In view of Fig. 1c, the KK scheme has a strong dependency on $N_c$.
Increasing $N_c$ from 100 to 500, ATC rates decreases dramatically, especially at the CWCs over
1.0 g m$^{-3}$. Unlike the KE scheme, ATC is allowable in the KK scheme even very low CWCs.
**2.4 Liu-Daum-McGraw-Wood (LD) scheme**
A generalized ATC parameterization was proposed by Liu et al. (2006). The approach
improved the representation of the threshold function by applying the expression for the critical
radius derived from the kinetic potential theory. The parameterization is given by
$$P_{ATC-LD}[\text{kg kg}^{-1}\,\text{s}^{-1}] = k b^6 q_c^3 N_c^{-1} \left\{1 - exp[-(1.03 \times 10^{16} N_c^{-3/2} q_c^2)^m]\right\}. \qquad (5)$$
Here, $\kappa (=1.1 \times 10^{10} \text{ kg}^{-2} \text{ m}^3 \text{ s}^{-1})$ is a constant. $\beta$ is a parameter related to relative dispersion $e$ of
cloud droplets, which is obtained from
$$b = \left[\frac{(1+3e^2)(1+4e^2)(1+5e^2)}{(1+e^2)(1+2e^2)}\right]^{\frac{1}{6}}. \qquad (6)$$
Here, a value of 0.5 is assigned to $e$ following Liu et al. (2006). The LD scheme is
characterized by the smooth transition in the vicinity of the ATC threshold.
**3 Description of the ensemble (EN) scheme**
As has been mentioned above, ATC rates predicted by different schemes can differ by
several orders of magnitude for a given CWC. Nowadays, it is still troublesome to judge which
scheme is preferred to others at all times (Ghosh and Jonas, 1999; Jing et al., 2019; Liu et al.,
2006; Michibata and Takemura, 2015). To the best of our knowledge, each one has its own
advantages and disadvantages. Keeping this fact in our mind, we propose a weighted the EN
scheme by employing the above-listed four commonly used ATC schemes, and the weighted





ensemble ATC rate ($P_{ATC\text{-}EN}$) is given by
$$P_{ATC-EN}[\mathrm{kg\,kg^{-1}\,s^{-1}}]=\frac{w_{KE}P_{ATC-KE}+w_{KK}P_{ATC-KK}+w_{LD}P_{ATC-LD}+w_{BR}P_{ATC-BR}}{w_{KE}+w_{KK}+w_{LD}+w_{BR}}.$$
(7)

Here, $w_{xx}$, referring to that for KE, KK, LD, and BR, respectively, is the weight of each ATC
scheme. It is worth noting that Eq. (7) is easy reduced into any single scheme form by setting all
$w_{xx}$ values of 0 except for one of them. Therefore, it is a flexible way to use any one or more
schemes to calculate $P_{ATC\text{-}EN}$ by adjusting $w_{xx}$. Of course, it is also convenient to reduce the
effect of any one of them by giving a small value of $w_{xx}$. At present, the same weights with the
value of 1.0 are assigned for all schemes for simplicity. Note that,the weights can be modulated
according to weather conditions. One of the features of the EN scheme is that the weighted
mean is calculated within a microphysics scheme, and the increasing of computation cost is
negligible.

Similar to an ensemble prediction system (Lewis, 2005), the EN scheme is expected to

reduce the potential uncertainties from the use of any ATC scheme alone under various CWC
conditions. For example, no cloud water converts into rain water in the KS scheme when the
cloud water is less than the threshold, while in the KK scheme it always occurs. However, the
KS scheme has much higher ATC rates owing to the linear relationship (Eq. 1), compared to
those of the KK scheme. Most importantly, the EN scheme is beneficial for the multi-scale
numerical weather and climate modeling systems, especially for variable resolution models
(e.g., the Model for Prediction Across Scales, MPAS (Skamarock et al., 2012), the
Global-to-Regional Integrated forecast SysTem, GRIST, (Zhang et al., 2019)), because it is
flexible to represent subgrid-scale cloud processes consistently across all model scales under
the various conditions. Depending on grid distance, one or more schemes can be used





independently in a variable resolution model. For example, we assign all $w_{xx}$ to 0 except for
$w_{KK}$ in fine grid distance region, and a mean value from the calculation of two or more
schemes is utilized in the grid distance transition zone.

To facilitate comparisons among the aforementioned ATC schemes, an idealized

experiment is performed with a wide range of CWCs in the calculations. A roughly value of $N_c$
is set to 300 cm$^{-3}$ in the continental clouds (e.g., Hong and Lim, 2006; Thompson et al., 2008).
For convenience, air density is approximately fixed at $1.29 \times 10^{-3}$ g cm$^{-3}$ here. It is noteworthy
that the value of 2 is assigned to $\mu$ for both BR and LD schemes. Figure 2 compares the EN
scheme with the selected four schemes with a wide range of CWCs from 0.01 to 1.0 g m$^{-3}$. One
can see that all the schemes yield ATC rates of $\sim 10^{-9}$ g cm$^{-3}$ s$^{-1}$, although there are significant
discrepancies among the different schemes. For the KS scheme, the ATC of cloud water to rain
water does not start until the CWC exceeds the threshold   $q_0$  (Eg. 1). In contrast, the other
schemes are allowable even given fair low CWCs.

Comparatively speaking, both KS and LD predicts larger ATC rate than the other ATC

schemes (the BR or KK scheme) for a given CWC. As for the former group, LD yields the
largest ATC rate with CWC below 0.6 g m$^{-3}$, while KS generates the largest ATC with CWC
over 0.6 g m$^{-3}$. Wood and Blossey (2005) argued that the ATC rate defined in LD would give
the total rate of mass coalescence among cloud droplets and is typically much larger than the
true ATC rate. With $N_c$ fixed at 300 cm$^{-3}$, the BR scheme shows close ATC rates to those of KK.
Note that the KK scheme, originally developed for the Large Eddy Simulation (LES) model,
yields the lowest ATC rate, followed by the BR scheme. The EN scheme provides a similar
pattern to LD, but nearly half ATC rates of those are yielded by the latter. It should be



emphasized that ATC rates are fairly sensitive to $N_c$ (Fig. 1), and a higher or lower $N_c$ would
cause greatly changes.

## 4 Simulations of an extreme rainfall event

### 4.1 Overview of the rainfall event

An extreme rainfall event hit Guangzhou megacity in the early morning hours of 7 May
2017. Within 18 hours (during the period of 2000 BST 6 May to 1400 BST 7 May), there
were 12 rain gauge stations over 250 mm during the rainfall process. The spatial distribution of
the rainfall appears two heavy rainfall cores over Jiulong (JL) and Huashan (HS) regions (Fig.
3a). The event was featured by the heaviest rainfall in Guangzhou megacity over the past six
decades with the maximum total amount of 542 mm within 18 hours at JL station (Fig. 3a). It
also broke the record of 3-h accumulated rainfall amount with the value of 382 mm. Another
marked feature of this rainfall event was its extreme hourly rainfall rate of 184 mm h$^{-1}$, which
is the second highest over the, Guangdong Province, China. The hourly rainfall rate is
comparable to the highest value of 188 mm h$^{-1}$ observed at Yangjiang station in Guangdong
Province on 23 June 2013.

### 4.2 Model configuration and experiment settings

This event was well simulated and investigated by Yin et al. (2020), focusing on the
effects of urbanization and orography. The WRF model configurations, and initial and
boundary conditions are the same as Yin et al. (2020) except for updating to the
WRF-ARW(v4.1.3) model (Skamarock et al., 2019) with several minor bugs fixed. For
convenience, an overview of the WRF model configures is presented here. The triple nested



domains have x, y dimensions of 313×202, 571×334, and 862×541 with grid sizes of 12, 4,
and 1.33 km, respectively. The WRF model physics schemes are configured with the
Thompson microphysics scheme (Thompson et al., 2008) with the modifications of ATC
parameterization, the rapid radiative transfer model (rrtm) (Mlawer et al., 1997) for both
shortwave and longwave radiative flux calculations, the Yonsei University (YSU) planetary
boundary layer (PBL) scheme (Hong et al., 2006), the MM5 Monin-Obukhov scheme for the
surface layer (Janjić, 1994), and the Noah-MP land-surface scheme (Niu et al., 2011). The Kain
cumulus parameterization scheme (Kain, 2004) is utilized for the outer two coarse resolution
domains, but being bypassed in the finest domain. All the three nested domains of the WRF
model are integrated for 18 hours, starting from 2000 BST 06 May 2017, with outputs at 6-min
intervals. The initial and outermost boundary conditions are interpolated from the National
Centers for Environmental Prediction (NCEP) Global Forecast System 0.25 degree re-analysis
data at 6-h intervals. In order to introduce realistically the UHI effects of the Guangzhou
metropolitan region, the Four-Dimension Data Assimilation (FDDA) functions are activated
(Reen, 2016) by performing both the surface observation nudging and the analysis nudging
from 2000 BST 6 to 0800 BST 7 May 2017. Please refer to Yin et al. (2020) for more details
about the model configuration.
As has been addressed above, it is convenient to a launch simulation with any of the above
listed ATC scheme alone. In total, two experiments were carried out with the EN and BR
schemes. It should be noted that the BR scheme was used originally in the Thompson scheme,
and the EN were newly coupled into the Thompson scheme in this work.



## 5. Results

### 5.1 Spatial distribution of accumulated rainfall

Figure 3 compares the spatial distribution of 18-h simulated total rainfall from the simulations with the EN and BR schemes to the observed. Generally speaking, both the schemes are able to capture main characteristics of the extreme rainfall event. One can see that the simulated rainfall amount compares favorably to the observed both at HS and at JL, although the JL storm has a 10-15 km eastward location shift. Yin et al. (2020) argued that the location errors may be related to large-scale meteorological conditions. Comparatively speaking, the EN and BR schemes performed better than others. The two centralized rainfall cores over HS and JL were successfully captured by the EN and BR schemes, with the simulated heaviest rainfall amount of 537 mm and 569 mm, respectively (Fig. 3b,c). As for the EN scheme (Fig. 3b), the simulated 18-h total rainfall were 320 mm and 537 mm over HS and JL, respectively, which was close to the observations of 341 mm and 542 mm (Fig. 3a). Similarly, the BR scheme performed equivalently to the EN scheme, with the maximum rainfall of 347 mm and 569 mm over Huashan and Jiulong regions, respectively (Fig. 3c). Note that the simulated heaviest over Huashan region were comparative among each other. In view of the results, we will compare the maximum hourly rainfall rates near JL from the simulations of the EN and BR schemes to that of observed in the next sections. It should be noted the results in the present study are a little better than (or equivalent to at least) those in Yin et al. (2020) because of the update of the WRF version4.1.3 model with some improvements in dynamical framwork and bug fixes.



### 5.2 Evolution of the simulated hourly rainfall


Figure 4 shows the observed and simulated time series of hourly maximum rainfall rates
over the Jiulong region. The observed peak rainfall near JL occurred at 0600 BST 7 May with
the hourly rates of 184 mm hr$^{-1}$. However, the simulated peak rainfall from the EN scheme took
place at 0700 BST 7 May, which was about 1 h later than the observed, with the hourly rates of
151 mm hr$^{-1}$. As for the BR scheme, the simulated peak rainfall rate occurred two hours later,
with the value of 144 mm hr$^{-1}$. As a matter of fact, both EN and BR schemes under-predicted
the peak hourly rainfall rate near JL. It is worthy to note that the observed timings of initiating
and ending of the ER production episode, i.e., near 0300 and 1000 BST 7 May, respectively,
were reproduced successfully. However, the both simulated peak rate occurred later than the
observed due to the slower increases in rain-producing rates than the observed. More
specifically, the observed hourly rate increased from about 16 mm hr$^{-1}$ to 184 mm hr$^{-1}$ just in
one hour (i.e., from 0500 to 0600 BST). However, the simulated from the EN scheme increased
from 0.3 mm hr$^{-1}$ at 0400 BST to about 79 mm hr$^{-1}$ at 0600 BST, and then to 151 mm hr$^{-1}$ at
0700 BST 7 May. As for the simulated with the BR scheme, it increased from 2 mm hr$^{-1}$ at 0400
BST to about 104 mm hr$^{-1}$ at 0700 BST, and then to 144 mm hr$^{-1}$ at 0800 BST 7 May. One
unique feature of the observed was the rapid increase of hourly rainfall rate. The rainfall
produced by the EN scheme peaked within 2 h while the BR scheme peaked over a period of 4
h. Additionally, both the simulated rainfall rates decrease over a period of several hours.
Generally speaking, the EN scheme performed much closer to the observed, compared to that
of the BR scheme. Note that the longer heavy rainfall period from the BR scheme contributed
partially to the over-prediction of the 18-h accumulated rainfall (Fig. 3c).



### 5.3 Evolutions of radar reflectivity

In view of the performance of the accumulated rainfall and the maximum hourly rainfall rates, we only compare the radar reflectivity from the simulations with the EN scheme to the results of the BR scheme. Figure 5 exhibits the structures and evolutions of convective cells over JL region by comparing the simulated composite radar reflectivity to the observed. The first well-organized radar echo formed near 0000 BST over Huashan region (not shown), which was located at the northern edge of a surface high-$\theta_e$ (equivalent potential temperature) tongue with significant convergence. As the southeasterly flow moved slowly eastward and the cold outflows resulted from previous convection, the Huashan storm dissipated while the storm began to develop over Jiulong region, both in its size and in intensity (Fig. 5a). The storm rapidly intensified during the period from 0430 to 0530 BST, with the peak reflectivity beyond 55 dBZ near the leading edge (Fig. 5a,b). The Jiulong storm moved fairly slowly, keeping more or less quasi-stationary shortly after its formation (Fig. 5a-c). Both the quasi-stationary nature and intense radar reflectivity explain the extreme rainfall production rate occurring at JL during the 1-h period of 0500 - 0600 BST. Subsequently, the Jiulong storm weakened, but its associated peak radar reflectivity still remained over 50 dBZ, which was consistent with the continued generation of significant rainfall near JL until 0800 BST (Fig. 4).

It is obvious that the both the EN and BR schemes captured the development of the Jiulong storm, with the main features that were similar to the observed, including quasi-stationary nature, southeastward expansion, and concentrated strong radar reflectivity during the extreme rainfall stage. Both simulations successfully generated a lower-$\theta_e$ pool with a distinct outflow boundary interacting with the moist southeasterly flow near the ground. It





should be noted that the initiation and organization of the both simulated Jiulong storm were
about 1.7 h later than the observed, and it occurred at a location nearly 10-15 km kilometers to
the east of the observed one. Generally speaking, both simulations with the EN and BR
schemes produced extreme rainfall amounts close to those observed and their spatial
distributions agree well with observations.

In terms of the spatial distribution of radar reflectivity, similar patterns can be seen

between the EN and BR schemes in the early stage before 0712 UTC, while differences are
visible at the extreme rainfall stage (Fig. 5e,h). One can find that the Jiulong storm simulated
with the EN scheme (Fig. 5f) developed more rapidly than that from the BR scheme, almost 1
h earlier than the latter (Fig. 5i). This was consistent with the timing lag in the hourly extreme
rainfall production (Fig. 4). Clearly, ACT process has an important influence on convective
development of deep convection associated with the extreme rainfall producing within the
Jiulong storm, which will be explored in view of the cloud microphysical processes in the
next section.
**5.4 The Effects on Macro- and Micro-physical Processes**

The spatial distribution of hourly rainfall, and temporal-averaged surface temperature

and horizontal wind during the period from 0600 BST to 0700 BST from the simulations with
the EN and BR schemes are displayed in Fig. 6. As has been stated above, the total rainfall
show slight difference between EN and BR over Jiulong region (Fig. 3b,c). In view of the
spatial distribution of the maximum hourly rainfall (Fig. 6), the EN scheme generated larger
rainfall area and stronger rainfall rate than those of the BR scheme, although both scheme
produced similar spatial distribution patterns in rainfall area, and temporal-averaged surface



temperature and horizontal wind filed. The result was consistent with the idealized experiment
in Fig. 2. For a given CWC, the EN scheme had a larger ATC rate, compared to the BR
scheme, and the difference becomes obvious with the increasing of CWC. The ATC process
mostly occurred at lower levels, resulting in higher number concentration of small raindrops
(Duan et al., 2020). The higher number concentration of middle-size raindrop was favorable
for coalescence of large precipitation particles from the upper levels, which made the larger
contribution to the extreme rainfall rate (e.g., Bao et al., 2019). As a result, the EN scheme
produced larger rainfall than the BR scheme. The result was consistent with Fu and Lin
(2019). The temporal and spatial extent of the "vigorous rain formation region" where most of
the rain was produced. Those features can also be viewed from the vertical sections in Fig. 7.
One can see that the largest radar reflectivity reaches the ground, like a bell on the ground
(Fig. 7a). This unique feature was reported by Li et al. (2020) based on the observations from
the S-band dual-polarization radar at Guangzhou station, Guangdong Province, China. The
bell-shaped radar reflectivity was consistent with the episode of the extreme hourly rainfall.
The strong radar reflectivity mainly resulted from raindrops coalescence owing to the higher
number concentration raindrop in the lower levels (Bao et al., 2020). That is to say, collecting
rain water by collision-coalescence process at the lower levels helped creat the large rainfall
rate at the ground. As for the BR scheme (Fig. 7b), a middle-level radar reflectivity cores was
obvious above nearly 1 km up to 4 km, indicating that raindrops coalescence occurred
intensively between those levels and evaporation of raindrop was significant below 1 km. The
evaporation near above the surface was a considerable factor abating the surface rainfall rate.
In view of the vertical distribution of radar reflectivity, the EN scheme generated a



maritime-like convective storm, whereas the convective storm simulated by the BR scheme
was close to a continental-like convection. That is to say, the latter have a smaller number of
raindrops near the surface.

Both the EN and BR schemes provide tilted storms in view of vertical cross from south to

north through the extreme rainfall. During this episode, the updraft was dominant in the storm,
and very weak downdraft occurred in the lower levels at the back of the convective storm.
Besides, both EN and BR reproduced very close thermal patterns in terms of potential
temperature. Note that the EN scheme had a slightly weaker in updraft than that of the BR
scheme, although only make the modification in the ATC parameterization in the
microphysics scheme (Fig. 7a,b), suggesting that change in cloud microphysical processes can
lead to some variations in dynamical processes.

The difference between EN scheme and BR scheme in updraft can be also viewed from

the cumulative contoured frequency by altitude diagrams (CCFAD) given in Fig. 8. CCFAD
presents the percentage of horizontal grid points with vertical motion weaker than the abscissa
scaled value for a given height (Yuter and Houze, 1995). In this study, vertical speeds are
binned with intervals of 1 m s$^{-1}$ based on the evelen model outputs with six-minute intervals
during the severe rainfall episode from 0600 BST to 0700 BST 7 May, 2017. Generally
speaking, the EN scheme shows similar CCFAD patterns to those of the BR scheme. However,
there are still various differences in the vertical motion. One can see there was a slight weaker
core but lower in the EN scheme simulation, compared to those of the BR scheme. During the
severe rainfall episode, the EN scheme produced the largest updraft nearly 15 m s$^{-1}$ at 5 km
level, while that was about 16 m s$^{-1}$ at 6 km level given by the BR scheme. On the contrast,



updrafts below 6 m s$^{-1}$ occurred more frequently in EN than that in the BR scheme. Overall,
the EN scheme provided a larger updraft area but slight weaker in upward speed, compared to
those in BR scheme. This is why the EN scheme had a larger spatial distribution of rainfall
than that of the BR scheme (Fig. 6a,b). Note that both EN and BR schemes had a slight
difference in downdraft in vertical distribution and the downdraft was mainly located below 2
km, which were also visible in the vertical cross sections (Fig. 7a,b).

As has been noted above, both the EN and BR schemes produced very close dynamical

patterns except for updrafts. However, differences were remarkable in cloud microphysical
processes. Figure 9 compares the temporal evolution of hydrometeors between EN scheme
and BR scheme. One can see that the EN scheme (Fig. 9a-f) produced similar hydrometeors
patterns to those of the BR scheme (Fig.9g-i). Overall, graupel was dominant above the
melting layer, while rainwater was considerable below the melting layer. Previous studies
(Franklin et al., 2005; Krueger et al., 1995; McCumber et al., 1991; Yin et al., 2018) proposed
that graupel was dominant in the tropical and subtropical clouds owing to plentiful water
vapor. Overall, the EN scheme mainly increased rainwater content and graupel, while only
slight differences in cloud water, cloud ice, snow, and water vapor, compared with those of
the BR scheme (Fig. 9m-r).

In terms of the difference in rainwater and graupel between the EN and the BR schemes

(Fig. 9m-r), we find that the ATC rate of the EN scheme played an important role in the
development of deep convection. Compared to the BR scheme, the higher ATC rate of the EN
scheme quickly produced more considerable number of small precipitation-sized drops within
updrafts in moderate- and lower-levels, and more of the small size raindrops were lofted by


445 the updrafts above the 0°C level and subsequently were fed for ice processes. Within this

446 graupel coexisted with more small supercooled rainwater region, stronger riming occurred

447 between ice particles and the small size rain drops. Consequently, more of the small

448 supercooled raindrops were converted into graupel by ice cloud microphysical process such as

449 riming, leading to a more rapid graupel production. At the same time (Fig. 9q), more

450 supercooled raindrops froze becoming more graupel embryos since bigger raindrops freeze at

451 warmer temperatures than smaller cloud droplets, and continue to grow by riming and/or

452 other processes. Consequently, graupel was increased at high altitude (above the 0°C ) levels.

453 It is well known that bigger water drops freeze at warmer temperatures than small drops.

454 Therefore, partial the small raindrops froze into graupel and snow particles, which contributes

455 the increasement in graupel and snow. Generally, a graupel particle has a larger size than a

456 raindrop with a given mass. Therefore, the larger graupel particle can collect more particles as

457 they fall downward in the storm, which helped creat the surface heavy rainfall rate. One can

458 see that the graupel increased rapidly nearly 12 minutes after the appearance of increasing

459 supercooled rain (Fig. 9n). It should be noted we try to understand cloud mirophysical

460 processes in the extreme rainfall based on our knowledge at present, and thus a rigorous

461 validation is required by comparing hydrometeors sink and terms in a future study.

462  As the increased graupel passed by the melting level, they started to melt leading to

463 more raindrops. In view of the strong radar reflectivity near the surface in Fig. 7a, the

464 raindrops from upper levels grew rapidly by collecting raindrops in the lower levels. In this

465 way, the extreme rainfall rate was generated in such a more rapid and efficient approach,

466 compared those of the BR scheme. During this stage, the increased ATC rate was linked to



ice-phase processes and modified graupel fraction above the 0°C level. As has been
mentioned earlier, the increased ATC rate played a certain role in dynamical feedbacks, and
the degree of modulation of water vapor, cloud water, cloud ice, and snow by the increased
ATC rate was negligible. These findings indicate that increased ATC rate were important in
the extreme rainfall that involved ice-phase processes of graupel above the 0°C levels and
warm-rain processes of rain drop in the lower levels. To summarize, the higher ATC rate of
the EN scheme produced more small precipitation-sized drops, and some of the small size
raindrops were lofted by above the 0°C level. Consequently, more graupel were generated by
riming and freezing processes. The rapid production of graupel played significant roles in the
development of the extreme rainfall. Collision and coalescence processes between liquid
particles appeared to be the mechanism of radar reflectivity increment toward the surface
within the storm core region.
We proposed the influence mechanism of ATC rate on the extreme rainfall by comparing
the simulated results between the EN scheme and the BR scheme. However, there are still
some limitations to figure out the complete effects of the increasing ATC rate on
microphysical and dynamical processes at present because those processes are entangled with
complicated interactions. Therefore, a better choice is to separate the effects on each process
by conducting high-resolution simulations with a sophisticated model, such as the approach of
Grabowski (2014). Certainly, the best way is to perform offline testing based on in-situ
observations, as was done by Wood (2005). Keeping those issues in our mind, further work is
needed to address this question.





## 6 Conclusions and Discussion


In this study, we designed an ensemble (EN) approach to improving ATC process
description in the cloud microphysics schemes. One unique feature of the EN approach is that
the ATC rate is a mean value based on the calculations from the several widely used ATC
schemes. Similar to ensemble prediction, this approach is aimed to improve the representation
of the ATC rate in case it has been treated by using an ATC scheme alone in the cloud
microphysics schemes. At present, the four widely used ATC schemes are selected, including
Kessler (1969) scheme, Berry and Reinhardt (1974) scheme, Khairoutdinov and Kogan (2000)
scheme, and Liu et al. (2006) scheme. In the EN scheme, each scheme is assigned a weight (Eq.
7) in order to modulate the importance of them. Certainly, the EN scheme is easily reduced
into any single scheme by setting all $w_{xx}$ values of 0 except for one of them. It is also convenient
to reduce the effect of a scheme by giving a small value of $w_{xx}$, even remove the effect of a
scheme by assigning a value of weight to 0. Under this framework, the ATC rates from the EN
scheme are compared to those from each of the several commonly used schemes by ideal
experiments, and a series of simulations are carried out for a urban-induced extreme rainfall
event over Southern China by using the EN, KE, BR, KK, and LD schemes which have been
coupled into the Thompson scheme in the WRF model (Thompson et al., 2008) in this work.
The results show that the EN scheme provides better simulations, compared to those from any
single ATC scheme used alone.
In this study, the ensemble approach has been employed to represent the ATC process in
the Thompson cloud microphysics scheme, which shows some advantages for simulation of





the extreme rainfall event, occurred on 7 May 2017 over southern China. It is important to
acknowledge that the conclusions are drawn from just one case study, and have not been
validated under a wider range of conditions over the world. In the forthcoming studies,
systematic assessment of more heavy rainfall events is planned to better understand the
performance of the EN scheme. It should be noted that there are still some limitation to the
EN scheme in the present study. Although a large number of ATC schemes are available, most
among them are not employed as ensemble member. For example, the Franklin scheme
(Franklin, 2008) took the effect of turbulence on the ATC process into account, which plays
important role in precipitation development (Chandrakar et al., 2018; Seifert et al., 2010).
Furthermore, equal weights were used in the present study for convenience. In other words, the
selected schemes have the same effect on the ATC rate. Moreover, only conventional
verifications were carried out, and the dependency of the performance of the ATC schemes on
model resolution was not considered in this study. A further examination with new approaches
( e.g., Wood, 2005; Grabowski, 2014) might provide important insights in the near future.
It is worth emphasizing that we focus our attention on the ATC from cloud water into
rainwater at present. Certainly, any source/sink term in a cloud microphysics scheme can be
dealt with the same method. Since developing a "unified" cloud scheme appears to be a
significant part of weather and climate model development in the coming years (Randall et al.,
2019), the EN approach may be a practicable way to reduce the potential uncertainty in cloud
and precipitation physical process, which will contribute to more accurate numerical model
development.



**Code and data availability**: The source code of the Weather Research and Forecasting model
(WRF v4.1.3) is available at https://github.com/wrf-model/WRF/releases (last access: July
2021). Modified WRF model codes and initial and boundary data used for the simulations are
available on Zenodo (https://doi.org/10.5281/zenodo.5052639). The National Centers for
Environmental Prediction (NCEP) Global Forecast System 0.25 degree re-analysis data at 6-h
intervals used for the initial and boundary conditions for the specific analysed period can be
downloaded at https://rda.ucar.edu/datasets/ds083.2/.
**Competing interests**: The author declares no competing interests.
**Author contributions.** J. Yin developed the weighted ensemble scheme and coupled the
scheme into the WRF model, with contributions from X. Liang. J. Yin tested and verified the
scheme with contributions from X. Liang, H. Wang, and H Xue. J. Yin wrote the manuscript,
and all the authors continuously discussed the results and contributed to the improvement of
the paper text.
**Acknowledgements**: This study is jointly supported by the National Natural Science
Foundation of China (42075083), National Key Research and Development Program of China
(2018YFC1507404 and 2017YFC1501806), and Development Foundation of Chinese
Academy of Meteorological Sciences (2019KJ026). The authors also acknowledge the use of
the NCAR Command Language (NCL) in the analysis of some of the WRF Model output and
the preparation of figures.



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



# **Figures**


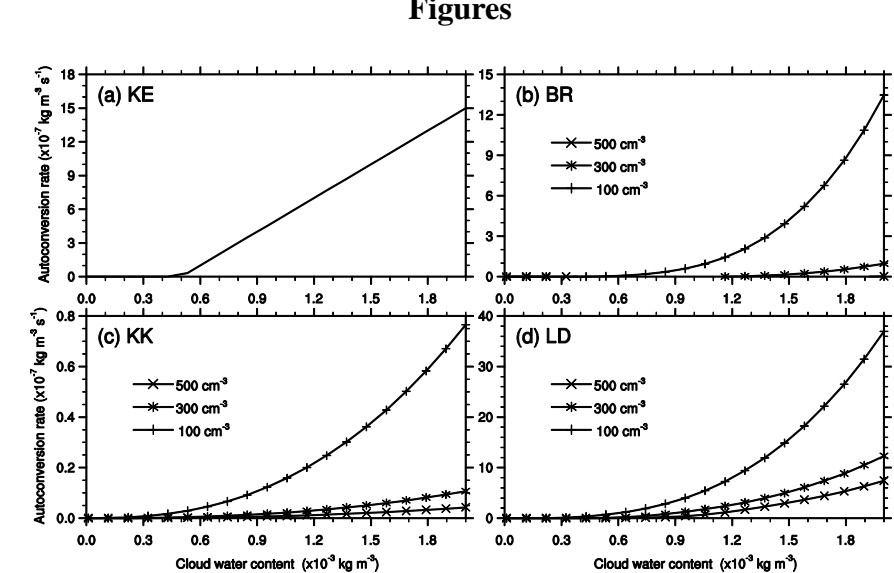


**Fig. 1** Evolution of autoconversion rates with a wide range of cloud water
content at given cloud number concentrations ($N_c$) of 100 cm$^{-3}$, 300 cm$^{-3}$, and 500
cm$^{-3}$, respectively. (a) KE denotes the Kessler scheme (1969), and (b) BR indicates
the Berry and Reinhardt scheme (1974); (c) KK and (d) LD represents the
Khairoutdinov and Kogan (2000) and Liu et al. (LD) schemes (2006), respectively.



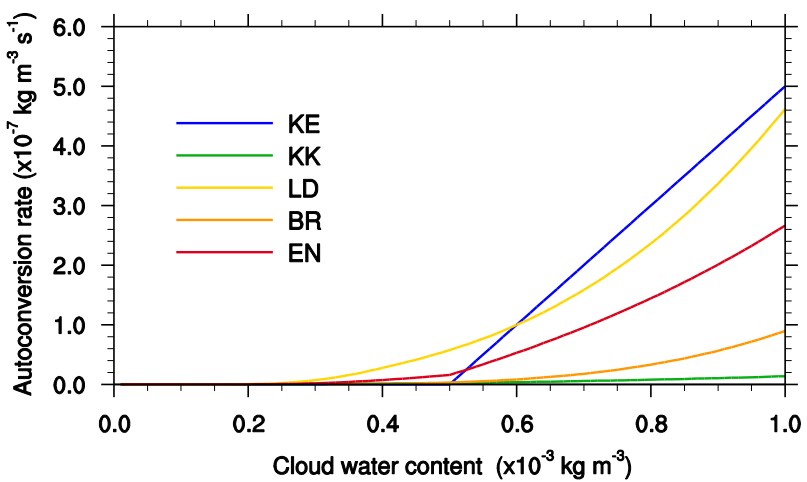

**Fig. 2** Comparisons of the EN scheme with the selected KE, BR, KK, and LD

schemes at a fixed $N_c$ of 300 cm$^{-3}$. (see text for further details)





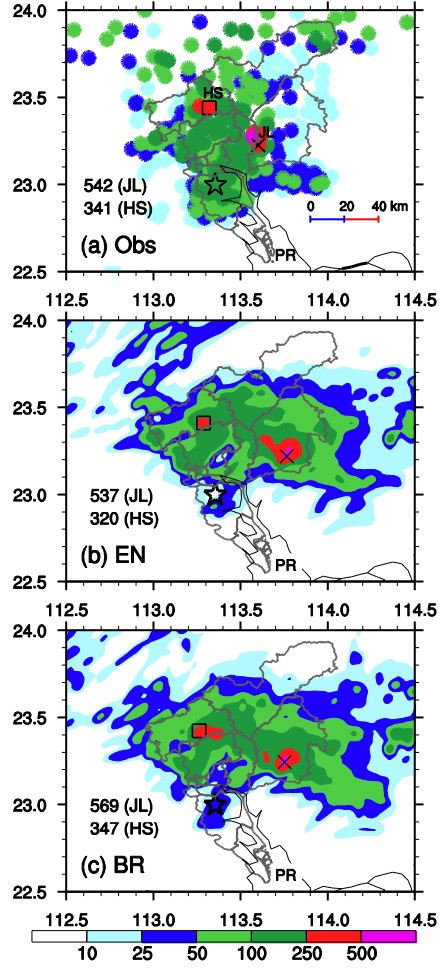


**Fig. 3** Spatial distribution of the 18-h accumulated rainfall during the period of 2000 BST 6 May to 1400 BST 7 May, 2017: (a) rain gauge observations and (b-c) simulations with the EN and BR autoconversion schemes. A cross sign (×) and a square sign (□) denote the locations where maximum hourly rainfall rates were (a) observed or (b-c) simulated near Jiulong (JL) and Huashan(HS), respectively. The values marked with JL and HS indicate the 18-h maximum accumulated rainfall amounts near the JL and HS, respectively. A star indicates the city center of Guangzhou, and the Pearl River is marked by PR; similarly for the rest of figures.



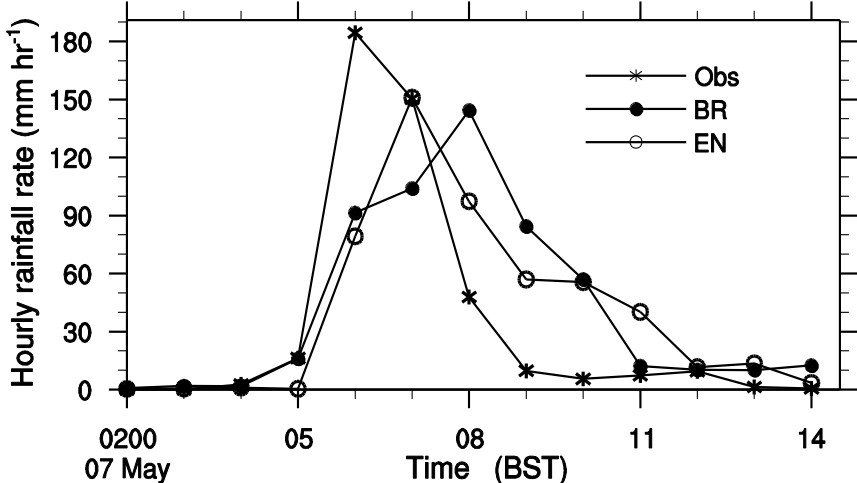

**Fig. 4** Time series of hourly rainfall rates (mm hr$^{-1}$) from rain gauge observations
(asterisks) and simulated with the EN scheme (circles) and the BR scheme (dots) near
Jiulong during the period of 2000 BST 6 - 1400 BST 7 May 2017. (see Fig. 3 for their
locations)

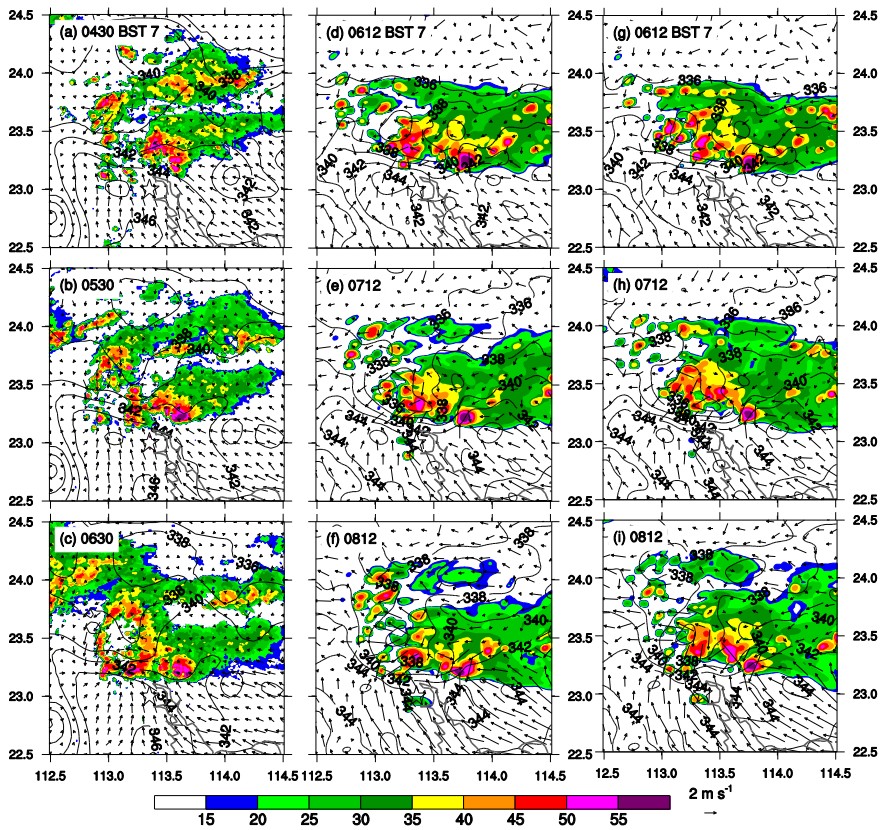

**Fig. 5** Horizontal maps of composite radar reflectivity (dBZ, shadings) and surface

(z = 10 m) horizontal wind vectors and equivalent potential temperature ($\theta_e$, contoured

at 2K intervals) during the extreme rainfall stage: (a-c) observed, (d-f) simulated with

the EN scheme, and (g-i) simulated with the BR scheme. A reference wind vector is

given beneath the right column next to the composite radar reflectivity color scale.

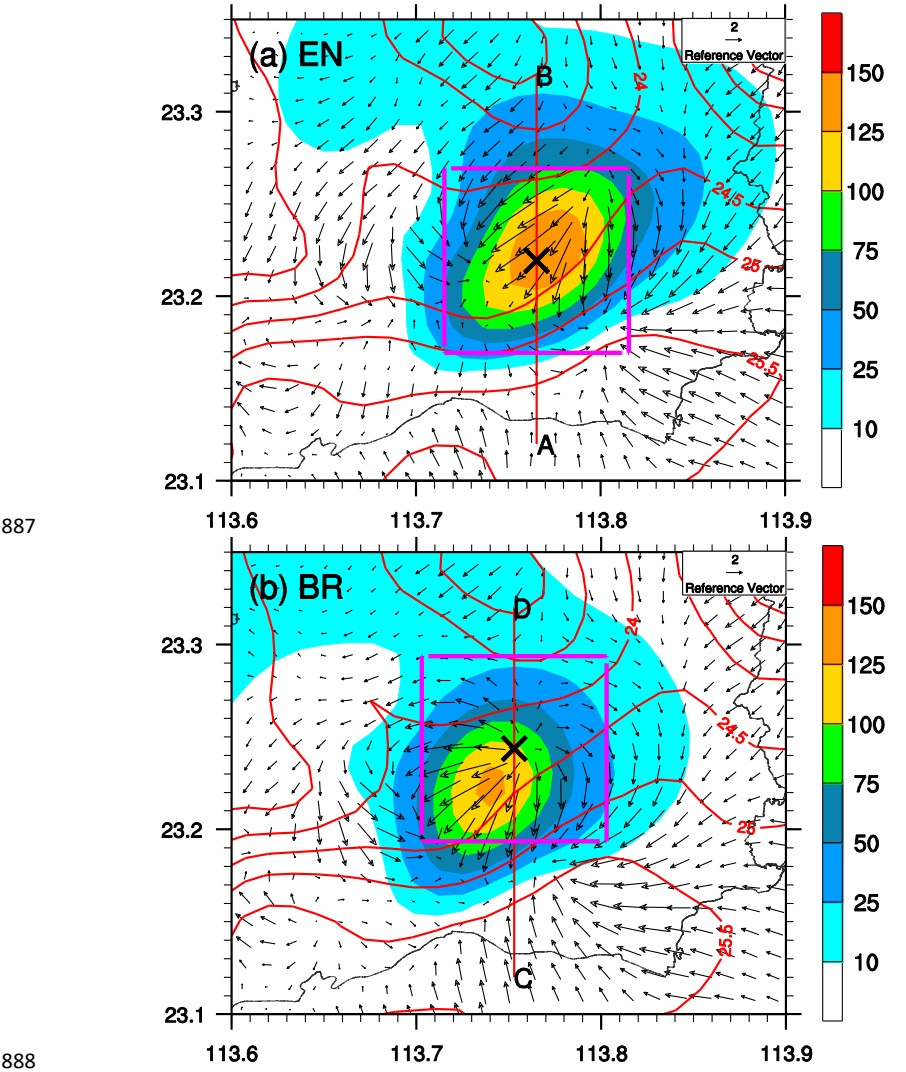



**Fig. 6** Spatial distribution of hourly rainfall amount (mm, shadings), temporal
-averaged surface temperature (contoured at 0.5℃ intervals) and horizontal wind
fields (vectors) during the period from 0600 BST to 0700 BST 7 May, 2017. The red
lines, A-B and C-D, indicate the locations of the vertical cross section in Fig. 7. The
two pink-squared boxes, covering an area of 0.1° × 0.1° with the center of the
maximum hourly rainfall, are marked for domain-averaged in Fig. 8 and Fig. 9.




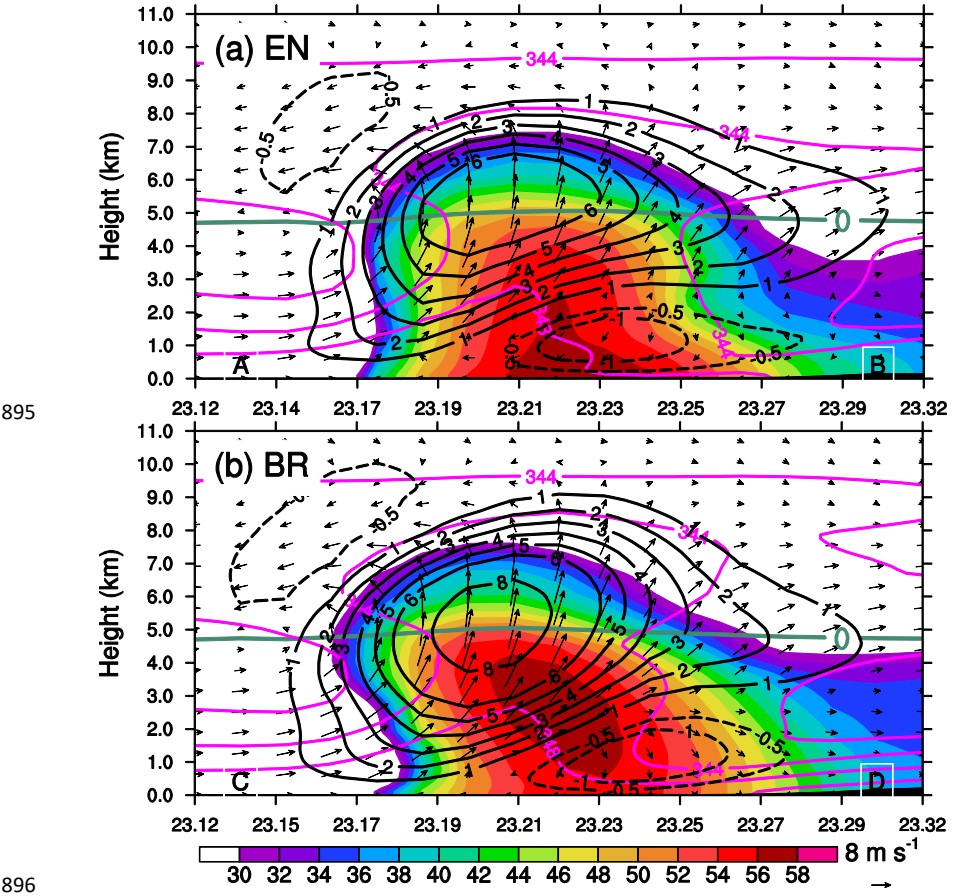


**Fig. 7** Temporal-averaged vertical cross sections along (a) A-B and (b) C-D in Fig.

6 of the simulated reflectivity (dBZ, shadings), vertical velocity (black contours, m s$^{-1}$),

in-plane flow vectors (vertical motion amplified by a factor of 2), and theta-e($\theta_e$,

pink-contoured at 4K intervals) during the period from 0600 BST to 0700 BST 7 May,

2017. Thick light green line indicates an isotherm of 0°C.

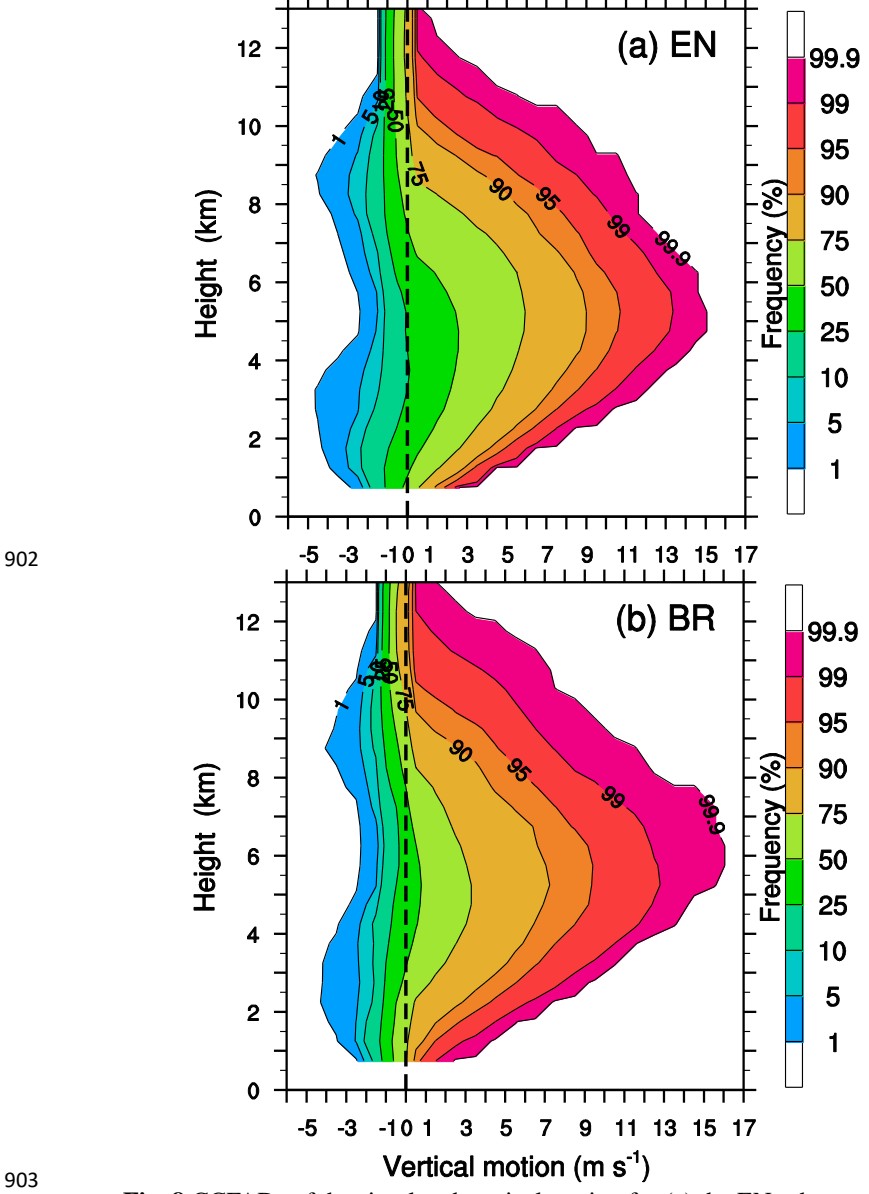



**Fig. 8** CCFADs of the simulated vertical motion for (a) the EN scheme and (b) the


BR scheme within the respective boxes marked with pink lines in Fig. 6. The CCFADs


are calculated from eleven model outputs with six-minute intervals during the severe


rainfall episode from 0600 BST to 0700 BST 7 May, 2017.




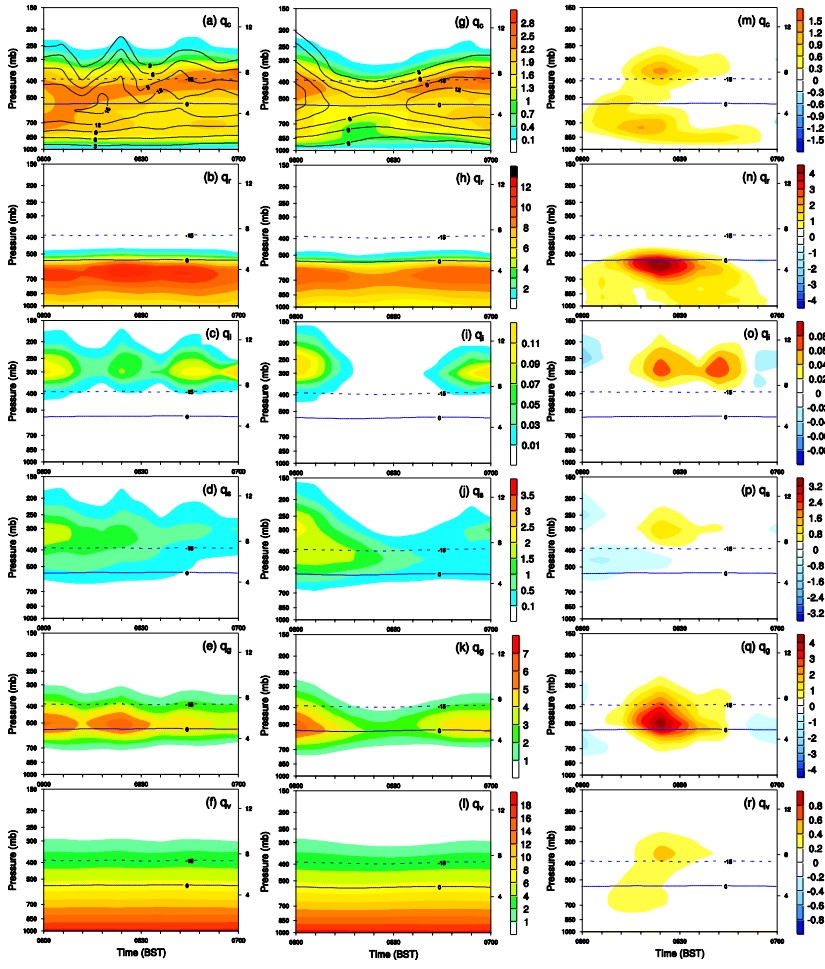

**Fig. 9** Comparison of time-height cross sections of domain-averaged mixing ratios between the EN scheme (a-f) and the BR scheme (g-i) during the period from 0600 BST to 0700 BST 7 May, 2017, within the domains marked with pink lines in Fig. 6. $q_c$, $q_r$, $q_i$, $q_s$, and $q_g$ denotes cloud water, rainwater, cloud ice, snow, and graupel, respectively. (m-r) gives the differences between EN and BR (i.e, EN – BR). Thick blue lines indicate isotherm of -15℃ and 0℃, respectively.