# Peer review of "Representation of the Autoconversion from Cloud to Rain Using a Weighted Ensemble Approach"

_Geoscientific Model Development, 2021_

## Author Comment (AC3)

**Response to Reviewer 2's comments**

*[General] Cloud microphysical processes are key components in parameterizing precipitation in numerical models yet large uncertainties remain between different autoconversion schemes. By combining four autoconversion rates schemes through a weight mean approach, the authors propose an ensemble scheme to try to avoid limitations of individual scheme. The ensemble scheme is then incorporated into the Thompson scheme to simulate an extreme rainfall event over Southern China. The rainfall extreme, distribution (both temporal and spatial) and hydrometer content are then compared with simulation with the Berry and Reinhardt (1974) scheme. Results show improvements in the timing and space of rainfall peak. This manuscript is well written, and the topic of this manuscript fits the scope of GMD. I recommend acceptance for publication after returning to the authors for minor revision.*

**Response:** Thank you very much for agreeing with us to the intention of this manuscript. We appreciate you for providing valuable comments and constructive remarks, which have helped improve our manuscript significantly

*[Major]*

The authors choose to compare simulation from EN with that from BR, I understand that it is partially because BR is used in the original Thompson scheme, but some results are kind of expected from Figure 2, for example, delayed rainfall peak. Did you compare the EN results with simulation using LD scheme?

**Response:** Yes. As has been addressed above, it is convenient to conduct a simulation with any of the above-listed schemes alone. In total, five experiments were carried out with the EN,KS,BR,KK, and LD schemes. The results indicate that the EN scheme provides better simulations than those treated by using any single scheme alone in terms of accumulated rainfall and extreme hourly rainfall rate.

Figure R1 compares the spatial distribution of 18-h simulated total rainfall from the simulations with the EN,KS,BR,KK and LD schemes to the observed. Generally speaking, all the schemes are able to capture the main characteristics of the extreme rainfall event. One can see that the simulated rainfall amount compares favorably to the observed both at HS and JL, although the JL storm has a 10-15 km eastward location shift. Comparatively speaking, the EN and BR schemes performed better than others. The two centralized rainfall cores over HS and JL were successfully captured by the EN and BR schemes, with the simulated heaviest rainfall amount of 537 mm and 569 mm, respectively (Fig. 1b,d). As for the EN scheme (Fig. R1b), the simulated 18-h total rainfalls were 320 mm and 537 mm over HS and JL, respectively, which was close to the observations of 341 mm and 542 mm (Fig. R1a). Similarly, the BR scheme performed similar to the EN scheme, with the maximum rainfall of 347 mm and 569 mm over Huashan and Jiulong regions, respectively (Fig.
R1d). One unique feature of the observations was the rapid increase in the hourly
rainfall rate. The rainfall produced by the EN scheme peaked within 2 h while the
BR scheme peaked over a period of 4 h. Both the simulated rainfall rates decrease
for several hours. Generally speaking, the EN scheme performed much closer to the
observed, compared to that of the BR scheme. Note that the longer heavy rainfall
period from the BR scheme contributed partially to the over-prediction of the 18-h
accumulated rainfall. In terms of the temporal evolution of radar reflectivity, one can
find that the Jiulong storm simulated with the EN scheme (Fig. 5f) developed more
rapidly than that from the BR scheme, almost 1 h earlier than the latter (Fig. 5i).
This was consistent with the timing lag in the hourly extreme rainfall production
(Fig. 4).
The heavy rainfall amounts over Jiulong region were underestimated by the KS, KK,
and LD schemes, with the heaviest rainfall amounts of 434 mm, 463 mm, and 473
mm, respectively (Fig. R1c,e,f). Note that the simulated heaviest over Huashan
region were comparative among each other.

[Figure]

**Fig. R1** Spatial distribution of the 18-h accumulated rainfall during the period from
2000 Beijing standard time (BST, BST = UTC + 8) 6 May to 1400 BST 7 May 2017.
(a) rain gauge observations, and (b-f) simulations with various autoconversion
schemes during the. A cross sign (×) and a square sign (□) denote the locations where
maximum hourly rainfall rates were (a) observed or (b-f) simulated near Jiulong (JL)
and Huashan(HS), respectively. The values marked with JL and HS indicate the 18-h
maximum accumulated rainfall amounts near the JL and HS, respectively. A star
indicates the city center of Guangzhou, and the Pearl River is marked by PR.

I appreciate the efforts of combining different schemes, but the manuscript lacks
descriptions and recommendations on how to adjust the weights in the EN when
simulating clouds in different synoptic systems, for example, continental deep
convection vs maritime drizzling stratocumulus. As the authors stated in Section 2
that each of the schemes spatializes in certain conditions. In the case demonstration,
if you adjust the weights to giving more weightings to schemes that are more
suitable for continental deep convection, will the results be closer to observations? It
might be too much work to add in this manuscript, but the EN scheme will be more
practically valuable if the authors can propose a recommending framework to adjust
the weights for different types of clouds.
**Response:** Thanks for your constructive comment. Adjusting the weights in the EN
scheme should give better results for different synoptic systems. At present, it is
troublesome to provide recommended weights for different synoptic systems
without a large number of tests and verification for specified weather conditions. In
this study, we focused on the EN approach and provided a flexible adjustment
interface for different aims. Users can adjust the weights according to their
objectives, even easily planting new members into the EN scheme. In order to help
users understand the options, a detailed description of the selected autoconversion
schemes (i.e., KE, BR, KK, and LD) has been added in the revised manuscript.
Keeping your suggestions in mind, a recommending framework to adjust the
weights for different types of clouds will be updated with the source codes on
Zenodo (https://doi.org/10.5281/zenodo.5052639) after detailed experiments in the
future.

*[Minor]*
*Line 99-100: please rephase this sentence. Do you mean the Cotton (1972) scheme*
*results in the peak cloud water content occur the earliest time, at the lowest cloud*
*attitude but has the lowest value as compared with other schemes?*
**Response:** Thank you very much for pointing this out. We have made revisions
accordingly.

*Line 119: remove are*
**Response:** Thank you very much for the reminder. Removed.

*Line 222-230: I do not get how the ensemble scheme can represent subgrid-scale*
*cloud processes with integrating one or more of the schemes over any assumed*
*CWC or Nc distributions like in Griffin and Larson, 2013. Any one of the four*
*schemes itself cannot represent subgrid-scale processes.*
**Response:** Not really. To the best of our knowledge, each individual scheme has its
own advantages and disadvantages, and there is no one scheme able to provide good
results at all times. For example, the LD scheme considering spectral dispersion was
more reliable for improving the understanding of the aerosol indirect effects, and the
KK scheme aimed at large-eddy simulation (LES). With the development of the variable resolution models, it is flexible to represent cloud processes consistently across all model scales under various conditions. Depending on grid distance, one or more schemes can be used independently in a variable resolution model. To avoid misunderstanding, the word "*subgrid-scale*" has been removed.

Line 288: …it is convenient to *conduct* a launch simulation…

**Response:** Thanks for your kind reminders. We revised the sentence as follows: "it is convenient to conduct a simulation…"

*Line 321: what is 'ER'? please elaborate when you first introduce an abbreviation.*

**Response:** ER denotes extreme rainfall. Corrected.

*Figure 7: is there radar observations at Jiulong site to compare reflectivity in observation and simulations? Does the observed maximum reflectivity extend to the surface?*

**Response:** The observed composite radar reflectivity was integrated by combining four individual radar observations at Guangzhou and its surroundings. Yes, the observed maximum stretched to the ground. Please refer to our previous observational analysis for detailed radar reflectivity vertical structures of the extreme rainfall, which is given in Li et al. (2020).

Li, M., Y. Luo, D. L. Zhang, M. Chen, C. Wu, J. Yin, and R. Ma, 2021: Analysis of a Record-Breaking Rainfall Event Associated With a Monsoon Coastal Megacity of South China Using Multisource Data. IEEE Transactions on Geoscience and Remote Sensing, 59, 6404-6414, doi:10.1109/TGRS.2020.3029831.

We appreciate you very much for your positive and constructive comments and suggestions on our manuscript, which are valuable in improving the quality of our manuscript.

---

## Author Response (AR1)

**Response to Reviewer 1's comments**

1 2

3 [General Comments] In the study, the authors explore the idea of improving 4 numerical simulation by improving the representation of the autoconversion from 5 cloud to rain (ACT) with a "weighted ensemble (EN)" ATC parameterization. To 6 construct the EN scheme, four widely used ATC parameterizations are employed, and then the EN scheme is coupled into the Thompson microphysics scheme in WRF. 7 8 With the EN scheme, the authors run nested (to  $\sim 1$  km) simulations of an extreme 9 precipitation event over southern China and then examine the results by comparison 10 of accumulated precipitation and radar reflectivity to observations. Besides, a 11 detailed analysis is given in vertical motion and hydrometeor mass mixing ratios. 12 The results show that the WRF model with EN run matches the observations better, 13 compared to the BR scheme which is used originally in the Thompson microphysics 14 scheme.

15 The premise of trying to improve cloud microphysical parameterization through such a kind of ensemble approach is interesting and potentially useful. One unique 16 17 feature of the ensemble approach is that the weighted mean is calculated within a 18 microphysics scheme with a negligible increase in computation cost. In my opinion, 19 the ensemble approach could easily be extended to other cloud microphysical 20 processes. Besides, the ensemble scheme appears to be a useful tool that can be used 21 to effectively switch between a single scheme alone as desired or to take the average 22 result of chosen ensemble members. This paper is generally in a good shape, well 23 organized, and conclusions well supported. However, there are a few items of 24 concern that the authors should address before being accepted for publication

**Response:** Thank you very much for your thorough review and constructivecomments that have helped improve the quality of our manuscript.

27

28 (1) Several grammar errors and typos throughout the text, please check carefully.

29 **Response:** We apologize for the language problems. We have revised the English 30 writing of the manuscript carefully. The errors of word choice, verb tense, sentence

31 structure as well as grammatical and bibliographical errors have been systematically

32 dealt with and the relevant mistakes have been corrected in the revised manuscript.

33 (a) Line 43 "articales" —>"articles"

34 Corrected.

35 (b) Line 51 "riandrops" —>"raindrops"

- 36 Corrected.
- 37 (c) Line 291 "were" —>"was"

- 38 Corrected.
- 39 (d) Line 512 suggest changing "more heavy" to "heavier"
- 40 Modified.
- 41 .....
- 42
- 43 (2) In Section 2, four widely used autoconversion schemes are employed in the
- 44 present study. Please elaborate on the advantages and disadvantages of these

45 schemes, which might tell readers more information.

- 46 **Response:** Thanks for your kind suggestion. Detailed descriptions about the selected
- 47 schemes have been added in the revised manuscript. For your convenience, the
- 48 revised portions are also given as follows.
- 49
- 50 For the Kessler (KE) scheme:

51 Kessler (1969) initially proposed a simple parameterization scheme that related the 52 autoconversion rate to cloud water content. Owing to the simple and linear expression, 53 the KE scheme is computationally straightforward to implement in numerical models. 54 However, the major limitation of the KE scheme results in its inability to identify 55 different conditions such as maritime and continental clouds (Ghosh and Jonas, 1999). 56 More specifically, the KE scheme only took cloud water content (CWC) into account, 57 while cloud number concentration was not incorporated. This may partially explain 58 the KE scheme yielded the large errors at low CWC proposed by Cotton (1972). 59 Besides, it is impossible to obtain the thresholds directly used in the scheme from 60 observations at present. However, cloud microphysical processes are sensitive to the 61 threshold (Plsselt et al., 2019). In order to get reasonable results, different values of  $q_0$ 62 were chosen by various studies. For instance, a value of  $0.5 \text{ g m}^{-3}$  is given in Kessler's 63 (1969), Reisner (1998), and Schultz (1995). Thompson (2004) reduced to a small 64 value of 0.35 g m-3. Kong and Yau (1997) and Tao and Simpson (1993) gave a value 65 of 2 g kg-1, while a small value of 0.7 g kg-1 was assigned in Chen and Sun (2002).

- 66
- 67 For the Berry-Reinhardt (BR) scheme

The BR scheme was developed theoretically in which not only CWC but also cloud number concentration was incorporated. An important characteristic is that maritime and continental clouds can be differentiated by the BR scheme using different parameters (Simpson and Wiggert, 1969; Pawlowska and Brenguier, 1996). Cotton (1972) argued that the BR scheme seems to underestimate rain formation in their simulations.

74

**75 For the Khairoutdinov-Kogan (KK) scheme**

The KK scheme was established based on a series of large-eddy simulations. The
 KK scheme uses a simple power-law expression based on bin microphysical

79 CWC and/or decreasing cloud number concentration. The simple expression is a key 80 advantage of the KK scheme, which makes it possible to analytically integrate the microphysical process rates over a probability density function (Griffin and Larson, 81 82 2013). In view of Fig. 1c, the KK scheme has a strong dependency on  $N_c$ . Increasing 83  $N_c$  from 100 to 500, ATC rates decrease dramatically, especially at the CWCs over 1.0 g m-3. Unlike other schemes, ATC is allowable in the KK scheme even with very 84 low CWCs, which might lead to overestimations under such conditions. 85 86 87 For the Liu-Daum-McGraw-Wood (LD) scheme 88 The LD scheme assumes that autoconversion rate is determined by CWC, cloud 89 number concentration, and relative dispersion of cloud droplets. Xie and Liu (2015) 90 suggested that the LD scheme considering spectral dispersion was more reliable for 91 improving the understanding of the aerosol indirect effects, compared to the KE and 92 BR schemes. 93 94 References: 95 Chen, S.-H. and Sun, W.-Y.: A One-dimensional Time Dependent Cloud Model, J. Meteor. Soc. 96 Japan, 80, 99-118, https://doi.org/10.2151/jmsj.80.99, 2002. 97 Cotton, W. R.: Numerical Simulation of Precipitation Development in Supercooled Cumuli-Part I, 98 Mon. Wea. Rev., 100, 757-763, 99 https://doi.org/10.1175/1520-0493(1972)100<0757:NSOPDI>2.3.CO;2, 1972. 100 Ghosh, S. and Jonas, P. R.: On the application of the classic Kessler and Berry schemes in Large 101 Eddy Simulation models with a particular emphasis on cloud autoconversion, the onset time of 102 precipitation and droplet evaporation, Ann. Geophys., 16, 628-637, 103 https://doi.org/10.1007/s00585-998-0628-2, 1999. 104 Griffin, B. M. and Larson, V. E.: Analytic upscaling of a local microphysics scheme. Part II: 105 Simulations, Quart. J. Royal Meteor. Soc., 139, 58-69, https://doi.org/10.1002/qj.1966, 2013. 106 Kessler, E.: On the Distribution and Continuity of Water Substance in Atmospheric Circulations, 107 Circulations. Meteor. Monogr., 10. American Meteorological Society, Boston, 1969. 108 Kong, F. and Yau, M. K.: An explicit approach to microphysics in MC2, Atmos.-Ocean, 35, 257-291, 109 https://doi.org/10.1080/07055900.1997.9649594, 1997. 110 Pawlowska, H., and J. L. Brenguier, A study of the microphysical structure of stratocumulus 111 clouds. Proc. 12th Int. Conf. Clouds and precipitation, Zurich, Ed. P. R. Jones, Published by

calculations. Generally, speaking, the autoconversion rate increases with increasing

112 Page Bros., Norwich, U.K., 123-126, 1996.

78

- 113 Posselt, D. J., He, F., Bukowski, J., and Reid, J. S.: On the Relative Sensitivity of a Tropical Deep
- Convective Storm to Changes in Environment and Cloud Microphysical Parameters, J. Atmos.
  Sci., 76, 1163-1185, https://doi.org/10.1175/JAS-D-18-0181.1, 2019.
- 116 Reisner, J., Rasmussen, R. M., and Bruintjes, R. T.: Explicit forecasting of supercooled liquid water
- 117 in winter storms using the MM5 mesoscale model, Quart. J. Roy. Meteor. Soc., 124, 1071-1107,

118 https://doi.org/10.1002/qj.49712454804 1998.

- Schultz, P.: An Explicit Cloud Physics Parameterization for Operational Numerical Weather
   Prediction, Mon. Wea. Rev., 123, 3331-3343,
- 121 https://doi.org/10.1175/1520-0493(1995)123<3331:AECPPF>2.0.CO;2, 1995.
- Simpson, j. and Wiggert, v.: Models of precipitating cumulus towers, Mon. Wea. Rev., 97, 471-489,
   https://doi.org/10.1175/1520-0493(1969)097<0471:MOPCT>2.3.CO;2, 1969.
- Tao, W.-K. and Simpson, J.: Goddard Cumulus Ensemble Model. Part I: Model Description, Terr.
   Atmos. Oceanic Sci., 4, 35-72, https://doi.org/10.3319/TAO.1993.4.1.35(A), 1993.
- Thompson, G., Rasmussen, R. M., and Manning, K.: Explicit Forecasts of Winter Precipitation
  Using an Improved Bulk Microphysics Scheme. Part I: Description and Sensitivity Analysis,
  Mon. Wea. Rev., 132, 519-542,
- 129 https://doi.org/10.1175/1520-0493(2004)132<0519:EFOWPU>2.0.CO;2, 2004.
- 130 Xie, X. and Liu, X.: Aerosol-cloud-precipitation interactions in WRF model: Sensitivity to
- 131 autoconversion parameterization, Journal of Meteorological Research, 29, 72-81,
- 132 10.1007/s13351-014-4065-8, 2015.
- 133

134 *(3) Line 377 "the EN scheme generated larger rainfall area and stronger rainfall*

- rate than those of the BR scheme". The result is interesting. I would suggest adding
  more explanation to make it easily understood.
- 137 **Response:** Given the spatial distribution of hourly rainfall during the period (i.e., 138 0600 BST to 0700 BST 7) when maximum hourly rainfall occurred, the EN scheme 139 generated larger rainfall area and stronger rainfall than those of the BR scheme, 140 although both schemes produced similar spatial distribution patterns in rainfall area, 141 and temporal-averaged surface temperature and horizontal wind filed. For a given 142 CWC, the EN scheme has a larger ATC rate, compared to the BR scheme, and the 143 difference becomes obvious with the increase of CWC. Consequently, the EN 144 scheme produced more rain water of small- to middle size, compared to the BR 145 scheme. The larger rain water was favorable for the coalescence of large 146 precipitation particles from the upper levels, which made the larger contribution to 147 the extreme rainfall rate. This is why the EN scheme produced larger rainfall than 148 the BR scheme.
- (4) Line 397-398 Evaporation does produce decreasing reflectivity field near the
  surface. However, large particle (raindrop) breakup is another microphysical
  process that can lead reflectivity values to decrease toward the surface.
- **Response:** Yes. Except for the evaporation, large particle (raindrop) breakup can lead reflectivity values to decrease toward the surface because reflectivity is much sensitive to raindrop size. In the present case, the evaporation of raindrops was remarkable. However, a slight difference was found in differential reflectivity Zdr in the lower levels (Fig. R1), indicating that large particle (raindrop) breakup was weak.

Fig. R1 Temporal-averaged vertical cross-section along C-D in Fig. 6 of the simulated differential reflectivity (dB, shadings) during the period from 0600 BST to 0700 BST 7 May, 2017.

- 161 (5) Line 402, The authors need to reword this sentence. It is hard to determine the
- 162 *raindrop number concentration.*

163 **Response:** Thank you very much for the reminder. We have removed the sentence.

(6) Although the ensemble approach is coupled in the WRF model, it might be
beneficial for a global modeling system with distinctly cloud microphysical
processes over the world. Some discussions in the last part may expand the
application scope of the ensemble approach.

168 **Response:** Thanks for your suggestion. We have extended this part with a detailed

169 discussion of the potential applications of the EN scheme.

170

- 171 We appreciate you very much for your positive and constructive comments and
- 172 suggestions on our manuscript, which are valuable in improving the quality of our 173 manuscript.

**Response to Reviewer 2's comments** 1 2 3 [General] Cloud microphysical processes are key components in parameterizing 4 precipitation in numerical models yet large uncertainties remain between different 5 autoconversion schemes. By combining four autoconversion rates schemes through 6 a weight mean approach, the authors propose an ensemble scheme to try to avoid 7 limitations of individual scheme. The ensemble scheme is then incorporated into the 8 Thompson scheme to simulate an extreme rainfall event over Southern China. The 9 rainfall extreme, distribution (both temporal and spatial) and hydrometer content 10 are then compared with simulation with the Berry and Reinhardt (1974) scheme. 11 Results show improvements in the timing and space of rainfall peak. This 12 manuscript is well written, and the topic of this manuscript fits the scope of GMD. I 13 recommend acceptance for publication after returning to the authors for minor 14 revision. 15 **Response:** Thank you very much for agreeing with us to the intention of this 16 manuscript. We appreciate you for providing valuable comments and constructive 17 remarks, which have helped improve our manuscript significantly 18 [Major] 19 The authors choose to compare simulation from EN with that from BR, I understand 20 that it is partially because BR is used in the original Thompson scheme, but some 21 results are kind of expected from Figure 2, for example, delayed rainfall peak. Did 22 you compare the EN results with simulation using LD scheme? 23 Response: Yes. As has been addressed above, it is convenient to conduct a**

**Response:** Yes. As has been addressed above, it is convenient to conduct a simulation with any of the above-listed schemes alone. In total, five experiments were carried out with the EN, KS, BR, KK, and LD schemes. The results indicate that the EN scheme provides better simulations than those treated by using any single scheme alone in terms of accumulated rainfall and extreme hourly rainfall rate.

29

30 Figure R1 compares the spatial distribution of 18-h simulated total rainfall from the

31 simulations with the EN, KS, BR, KK and LD schemes to the observed. Generally

- 32 speaking, all the schemes are able to capture the main characteristics of the extreme
- 33 rainfall event. One can see that the simulated rainfall amount compares favorably to
- 34 the observed both at HS and JL, although the JL storm has a 10-15 km eastward
- 35 location shift. Comparatively speaking, the EN and BR schemes performed better
- than others. The two centralized rainfall cores over HS and JL were successfully
- 37 captured by the EN and BR schemes, with the simulated heaviest rainfall amount of
- 38 537 mm and 569 mm, respectively (Fig. 1b,d). As for the EN scheme (Fig. R1b), the
- 39 simulated 18-h total rainfalls were 320 mm and 537 mm over HS and JL,
- 40 respectively, which was close to the observations of 341 mm and 542 mm (Fig. R1a).
- 41 Similarly, the BR scheme performed similar to the EN scheme, with the maximum

- 42 rainfall of 347 mm and 569 mm over Huashan and Jiulong regions, respectively (Fig.
- 43 R1d). One unique feature of the observations was the rapid increase in the hourly
- 44 rainfall rate. The rainfall produced by the EN scheme peaked within 2 h while the
- 45 BR scheme peaked over a period of 4 h. Both the simulated rainfall rates decrease
- 46 for several hours. Generally speaking, the EN scheme performed much closer to the
- 47 observed, compared to that of the BR scheme. Note that the longer heavy rainfall
- 48 period from the BR scheme contributed partially to the over-prediction of the 18-h
- 49 accumulated rainfall. In terms of the temporal evolution of radar reflectivity, one can
- 50 find that the Jiulong storm simulated with the EN scheme (Fig. 5f) developed more
- 51 rapidly than that from the BR scheme, almost 1 h earlier than the latter (Fig. 5i).
- 52 This was consistent with the timing lag in the hourly extreme rainfall production
- 53 (Fig. 4).
- 54 The heavy rainfall amounts over Jiulong region were underestimated by the KS, KK,
- and LD schemes, with the heaviest rainfall amounts of 434 mm, 463 mm, and 473
- 56 mm, respectively (Fig. R1c,e,f). Note that the simulated heaviest over Huashan
- 57 region were comparative among each other.

---

## Author Response (AR2)

**Response**

Dear Dr. Topping,

We highly appreciate you for reviewing our paper (entitled "*Representation of the Autoconversion from Cloud to Rain Using a Weighted Ensemble Approach*", GMD-2021-230) and providing valuable comments, which are valuable in improving the quality of our manuscript. We have gone through the entire manuscript and several minor grammatical mistakes were eliminated.

Looking forward to hearing from you soon.

Sincerely yours,

Dr. Jinfang Yin

yinjf@cma.gov.cn

State Key Laboratory of Severe Weather, Chinese Academy of Meteorological Sciences, Beijing 100081, China

December 15, 2021

Line 88 'As were noted ' please change to 'As noted..'

**Response:** Thank you very much for the reminder. Revised accordingly.

Except for Line 88, several minor grammatical mistakes were eliminated as follows.

Line 17 "improve"   —>   "improving"

Line 496 "to figure out"   —>   "in figuring out"